# Blood Work: Managing Menstruation, Menopause and Gynaecological Health Conditions in the Workplace

**DOI:** 10.3390/ijerph18041951

**Published:** 2021-02-17

**Authors:** Katherine Sang, Jen Remnant, Thomas Calvard, Katriona Myhill

**Affiliations:** 1Edinburgh Business School, Heriot Watt University, Edinburgh EH14 4AS, UK; j.remnant@hw.ac.uk (J.R.); k.myhill@hw.ac.uk (K.M.); 2Business School, University of Edinburgh, Edinburgh EH8 9JS, UK; thomas.calvard@ed.ac.uk

**Keywords:** blood, body work, gynaecological health, higher education, menstruation

## Abstract

The menstrual cycle remains neglected in explorations of public health, and entirely remiss in occupational health literature, despite being a problematic source of gendered inequalities at work. This paper proposes the new concept of blood work to explain the relationship between menstruation (and associated gynaecological health conditions) and employment for women and trans/non-binary people. We build on and extend health and organisational literature on managing bodies at work by arguing that those who experience menstruation face additional work or labour in the management of their own bodies through the menstrual cycle. We discuss how this additional labour replicates problematic elements that are identifiable in public health initiatives, in that it is individualised, requiring individual women and trans/non-binary people to navigate unsupportive workplaces. We present findings from an analysis of qualitative survey data that were completed by 627 participants working in higher education, revealing that employees’ blood work comprises distinct difficulties that are related to the management of painful, leaking bodies, access to facilities, stigma, and balancing workload. We suggest developing supportive workplaces and public health policies, which refocus the responsibility for accessible, equal workplaces that accommodate menstruating employees, and those with gynaecological health conditions.

## 1. Introduction

In 2016, a company in Bristol became the first organisation in the UK to launch “period leave” for its women employees [1], unusually situating menstruation as apart from sickness absence more generally. Such examples of changes in policy point to what is otherwise a strikingly absent topic in studies of occupational health and organisation studies on menstruation, conceptualised as one of the three Ms of gendered working lives (with the other two being maternity and menopause [2]). 

The current paper addresses this gap in understandings of how people manage menstruation and associated gynaecological health conditions in the workplace. We take a biopsychosocial approach to understanding experiences of menstruation, discussing public health literature on the issue and comparable topics, recognising menstruation as a distinctive issue inclusive of physical symptoms, stigma, shame, and social interactions, including those in the workplace. We explore conceptualisations of types of work that can be considered to share aspects with managing menstruation—including the disposal of waste and management of bodies at work. 

We present qualitative survey data that were collected from employees working in higher education settings (n = 627) and highlight key themes emerging from the data. We suggest that those who menstruate have an additional, distinctive form of physical and emotional labour to carry out, that of blood work. We argue that blood work is made of four key tenets that resemble wider issues relating to the management of health in public spaces: managing painful and leaky bodies, accessing adequate facilities, stigma, and managing workload. 

We conclude by discussing the additional labour, blood work, undertaken by those who experience menstruation, and associated gynaecological health conditions. We suggest a move from the current individualised responsibility of blood work, and conclude with recommendations for supportive workplace practices and policies to better accommodate the needs of employees managing their menstrual health and wellbeing. 

Although the authors of this article discuss people of any gender identity who experience menstruation, in discussing existing research we default to the language that is used in those studies.

## 2. Background

### 2.1. Understanding Menstrual Health at Work

Although there are some concerns regarding the medicalisation of menstruation within Western contexts [3], it has been argued that, globally, menstruation should be considered to be a public health issue [4]. Classifying menstruation, as such, has been in discussion of menstrual health practices in the Global South [4] but can also be understood as a recognition of the collection of physical symptoms and associated gynaecological health conditions women, girls and trans/non-binary individuals experience. Menstruation shares elements with other public health concerns such as managing stigma [5] which is present in public health discussions of obesity [6] and mental health [7] as well as having health implications in and of itself [8]. Health studies from across the world show that menstruation and menstrual health problems can be associated with reduced quality of life, negative employment outcomes, and symptoms, such as fatigue, which pose challenges to workplace engagement [9,10]. 

Little is known regarding the impact of menstruation on people’s careers [11,12]. Examining a cohort of menstruating women in the U.S., Cote et al. [13] demonstrated that heavier menstrual flow is associated with loss of work, resulting in significant negative financial implications for women. These effects may be more pronounced for those who experience additional gynaecological health conditions. Endometriosis has been revealed to be associated with increased sickness absence among Danish workers, and the long delay between onset of symptoms and diagnosis is linked to low work ability [14]. Another study of Korean workers identified that irregular menstruation is associated with greater part-time employment and unemployment [15].

In order to fully explore these experiences of menstruation, it is crucial to understand how they are socially situated [16] and how they are managed in the workplace. Despite the increased presence of women in the workplace over the 20th Century and general interest in wellness at work, research exploring the management of menstruation remains decidedly focused on arenas that are outside of the workplace [17]. This important research explores cultural associations between menstruating and impurity and the resulting limited access to public spaces for women and girls [18]. In low to middle income countries, it has become apparent that data on how women manage menstruation and work are very sparse, being complicated by women’s predominant employment in informal sectors where there are fewer legal protections [19]. University based work offers excellent insight into the management of menstruation in white collar jobs where there is a perception of work flexibility, an area under explored within wider literature on menstruation and employment.

In higher income countries work is emerging that starts to examine women’s experiences of breastfeeding and menopause at work more directly [20]. Jack et al [21], in a study of women working in higher education in Australia, identified important temporal considerations in how women experience menopause at work and their related sense of agency. Women reported shared experiences around menopause, while the ‘misbehaviour’ of their bodies (its leakiness at times) was a direct challenge to the unspoken male-body norm that underpins notions of workers. This implicit norm of the male body [22] affects those who are menstruating, as they may try to mask their symptoms of menstruation to continue working without disruption [23], “passing” as non-menstruating [24].

In the workplace more broadly, wellbeing has become a central tenet of human resource management (HRM), although this usually focuses on individuals’ psychological wellbeing, in terms of stress or resilience [25], with physical health dimensions and more structural inequalities remaining neglected [26] or specifically focused on the wider health outcomes that are associated with precarious work or worklessness [27], rather than within individual sectors or organisations. A focus on productivity in organisations can have embodied effects that fall disproportionately on those whose bodies do not fit bureaucratic regimes, especially women and trans/non-binary people during menopause, menstruation, and pregnancy [3]. 

Current data suggest that menstrual health problems are associated with economic hardship for women and trans/non-binary people. However, the relationship is not likely to be straightforward, but to involve reciprocal, interactive effects. It might be the case that individuals who are working experience less severe menopausal symptoms [28]. Social scientists can be reluctant to explore menstruation from a medical perspective, voicing concerns that doing so extends patriarchal understandings of the female body as a ‘problem’ in the workplace [21]. However, this view could potentially undermine the somatic symptoms of menstruation and associated gynaecological health conditions women and trans/non-binary people experience. This lack of academic exploration is replicated in public health research, focused as it is on experiences of menstruation outside of the workplace, while occupational health research has yet to meaningfully explore experiences of menstruation. This paper addresses this by explicitly exploring the management of menstruation and associated gynaecological health conditions in the workplace. In the following section, we reflect on the workplace and workplace practices that provide additional insight into the management of bodies and bodily processes at work. 

### 2.2. The Body at Work and (Dirty) Body Work

There is increasing academic interest in the corporeal body at work, and how physiological, embodied experiences inform relationships and interpretations in workplaces [29,30,31]. This interest in organisational embodiment includes specific gendered concerns around experiences, such as maternity and menopause [21,32], but it is largely remiss in discussing menstruation. Building on and extending this literature involves considering theoretical perspectives of how people perceive, manage, and exploit bodies at work. Here, we focus particularly on approaches that emphasise the aesthetic, stigmatizing, or ‘dirty’ conditions necessitating bodily work, given the powerful socializing and repressing effects in many societies that reinforce the stigmatization of menstrual blood [5,23,33]. 

After many years of exclusion, the body has come into focus as a key material aspect of how labour is organised, affecting the lived experience of work for specific workers (e.g., [21,32,34,35,36]). The concept of “body work”, for example, focuses on “work which people are expected to do to their *own* bodies, especially in the context of meeting the expectations of their employers or peers... [and] the experience of those whose paid work involves the care, adornment, pleasure, discipline and cure of others’ bodies” ([37], p. 497). Workers in health and social care have both direct and indirect contact with the bodies of others in the workplace; their work is affected by definitions of bodies, divisions of labour, and specific social interactions, such as touching the live bodies of patients, but also managing the disposal of bodily fluids, such as blood [37,38]. As a concept, body work can also be used to understand the labour of individuals in managing their own bodies, health/wellness, and appearance [39]. However, body work is distinct from related embodiment concepts, like aesthetic labour, which concerns how organisations mobilise, develop, and commodify employee bodies to improve profitability, typically in the service economy [40,41]. 

Body work in the sense of managing one’s own body has important temporal, spatial, and gendered aspects. Temporally and spatially, in the sense that the body must be attended to differently in particular moments, stages of life, relationships, and locations, and cannot be postponed or managed to a time or place more convenient for the worker [21,42]. Gendered, in the sense that body work can involve considerable intimacy when working with women’s reproductive systems and the social shame that can surround breasts, milk, blood, vulvas, and vaginas [23,32,43,44]. Thus, women experience pressure to undertake work to preserve masculine organisational norms of a disembodied, ‘ideal’ worker [45,46,47]. 

The concept of ‘dirty work’ can be married up with body work, if we consider how dirty work has been used to describe work that has an ‘occupational taint’ due to the physical substances and materials that workers encounter, as well as their moral, social and emotional connotations [48]. Examples of gendered forms of dirty work that have been explored include sex shop workers [49], gynaecological nurses [50], domestic workers [51], and care workers [52]. Among the workforces dominated by men, physically dirty forms of work studied also include butchery [53], refuse collection, and waste management [54]. Dirty work often has very specific health risks that are attached to it that relate to contamination relating to bodily fluids [55,56]. 

Elements of dirty work that can be applied to menstrual body work include exposure to blood, stigmatised body parts, the management of waste materials, and, to some extent, pain. However, interdisciplinary research into menstruation and menstrual issues has generally not conceptualised menstruation in terms of either body work or dirty work. Research into menstrual refuse, for example, has not reflected on workers’ managing their own bodies, instead focusing on the production, marketing, and distribution of menstrual product bins in which women can dispose of bloodied products [57,58]. However, women and trans/non-binary people also face difficult decisions about how to manage menstruation, where they will be seen doing so in public spaces and organisations [59,60]. Thus, organisational structures and environments affect these individuals’ options for managing their bodies and menstruation. Menstrual body work is captured. to some extent, by the construct of ‘menstrual etiquette’, which reflects the double burden of having to discursively and practically render menstruation invisible without proper social or infrastructural support, and without being able to call attention to its absence for fear of breaching silence on the topic [60]. 

In many environments, products and services are designed in ways that are orientated toward men and male bodies [61], with menstruation and gynaecological health, as material realities of women [62] and trans/non-binary people’s bodies having remained absent from discourses of the body at work. The absence of the voices of those who menstruate is a form of silence at work, where what is not said is as important as what is said [63]. Indeed, the cultural code of menstruation has generally been one of silence and concealment, compounded by an internalised taboo and feelings of shame and disgust. It is from these corporeal and affective conditions that the need for menstrual body work arises [64]. 

### 2.3. Women in Academia

Women are still under-represented in senior positions in universities and they face a range of difficulties in their day-to-day working lives, including a ‘chilly’ climate of discrimination and sexual harassment [65,66]. Academia can be characterised by long working hours, intensive work, and rigid notions of an ‘ideal worker’, who dedicates themselves entirely to work [47]. The ways in which work is allocated in universities can act to hinder women’s career progression [67]. For example, women are more likely to be allocated pastoral care and teaching responsibilities, which are not rewarded within academic promotion structures. While male academics may work over their contractual hours by choice to undertake research, women academics are forced to work extra hours due to the teaching and administrative work that they are given [68]. 

Even where women academics are research active, they may find their research denigrated within their universities, especially by colleagues [69]. Women academics are often excluded from key networks within academia that generate high-quality research outputs that are essential for career progression [70]. These pressures have evidenced negative health outcomes for women academics, including stress and burnout [71]. Despite the range of research documenting the difficulties that are faced by women academics, menstruation or menopause have yet to feature in these discussions. We conceptualise gendered aspects of managing the body at work by considering menstrual health at work. We propose the concept of blood work, which we define here as the individualised labour that is undertaken by those who experience menstruation and gynaecological health conditions to ‘manage’ their bodies to fit professional norms that were established by those who do not experience menstruation. 

## 3. Method

### 3.1. Study Design

This study followed a survey design, using an online survey tool with open-ended questions to elicit qualitative responses in 2017. The aim of the study was to understand the lived experiences of those who experience menstruation, menopause, and gynaecological health conditions while working in higher education, in order to extend gendered understandings of the body at work. This allowed for data to be collected from a larger sample of participants than an interview-based approach would have allowed for, and enabled participants to complete data collection at their own pace. Online surveys are established tools for collecting data from participants living with chronic health conditions [72], despite limitations relating to access and potential brevity in responses [73]. Research exploring workplace experiences of menstruation and menopause have been largely interview based to access more detailed data from fewer participants [1,21].

### 3.2. Participants

The final sample consisted of 636 responses to the survey. There were 627 usable responses after removing incomplete and repeated responses. Most of the participants worked within UK universities (80%), with the remaining 20% either having previously worked in UK academia or working in universities outside of the UK. For the purposes of the study, undertaking a PhD was also considered as working within academia. Over 95% of respondents worked in academic roles from PhD student through to Professor, with the remaining 5% working in Professional Service roles, including librarian, estates manager, and clinical trials manager. In terms of employment status, 56% of respondents were on open-ended/permanent contracts, either full-time (48%) or part-time (8%). An additional 21% were on full- or part-time temporary contracts, and the remaining 21% were studying PhDs either full- or part-time, with less than 2% not currently in work or study. Despite the gendered forms of management in university settings being well understood (21), little is known regarding employees’ embodied experiences of working in Higher Education. Universities offer excellent insight into the management of menstruation in white collar jobs, where there is a perception of work flexibility. 

This sample is broadly consistent with recent data on UK academics, although the estimated prevalence of staff on precarious contracts can vary [74]. Of those working within academic roles, most of the participants worked in social sciences and humanities (81%), including law, management studies, education, and fine arts. The remaining 19% worked in science, technology, engineering, mathematics, and medicine (STEMM) subjects. In terms of national backgrounds, 82% of respondents were based within the UK, with the remainder living in the EU or Australia, South Africa, and North America. In terms of gender, 99% of respondents identified as cis-gender women, with 11 participants identifying as queer, agender, non-binary, or as men. Participants’ ages ranged from early 20s to 65, with most participants aged between 25 and 45. 

### 3.3. Data Collection

The online survey asked participants to describe their experiences of menstruation, menopause, and gynaecological health conditions at work. Open-ended questions were asked in relation to managing menstruation, menopause, and gynaecological health conditions at work, including the accessibility of facilities, the impact of managing menstruation in relation to teaching, research, administration, and conferences, taking leave, access to sanitary products, and ability to discuss menstruation and gynaecological health at work. These questions gave respondents the opportunity to describe the work that they engaged in to manage their menstrual and gynaecological health at work. Finally, the participants were also asked to describe their gender, job title, discipline, location of work, and contract type (e.g., full-time; open-ended).

The survey was distributed via social media in order to achieve a wide range of responses, resulting in a convenience sample. As such, we make no direct claims of generalisability. Rather, we intend to explore varied experiences of menstruation, menopause, and gynaecological health conditions, and their management in the workplace. The study received full ethical approval from the lead author’s institution. The survey respondents were informed of their right to withdraw, and that their responses would be anonymised and not shared with a third party. As such, the participants were fully informed of the purpose of the study. 

### 3.4. Data Analysis

The open-ended responses of the survey were subjected to qualitative thematic analysis by adopting the six-step analytical approach that was advocated by Braun and Clarke [75]:Familiarising yourself with your data: transcribing data (if necessary), reading and rereading the data, noting down initial ideas.Generating initial codes: coding interesting features of the data in a systematic fashion across the entire data set, collating data relevant to each code.Searching for themes: collating codes into potential themes, gathering all data relevant to each potential theme.Reviewing themes: checking the themes work in relation to the coded extracts (Level 1) and the entire data set (Level 2), generating a thematic map of the analysis.Defining and naming themes: ongoing analysis to refine the specifics of each theme, and the overall story the analysis tells; generating clear definitions and names for each theme.Producing the report: the final opportunity for analysis. Selection of vivid, compelling extract examples, final analysis of selected extracts, relating back of the analysis to the research question and literature and producing a scholarly report of the analysis.

The coding process was deductively driven by a concern for identifying themes that resonated with gendered inequalities in the workplace and the conceptual framing that managing menstrual and gynaecological health involved emotional and physical labour. The coding process was also driven inductively, in terms of the specific emotions and explanations being given for inequalities and blood work, and respondents’ accounts of their distinctive health conditions, problems with access to facilities, and employment accommodations that they felt had an impact on their working lives.

## 4. Findings

Most of the respondents experienced menstruation (84%), with 17% currently or previously having experienced menopause (several participants were currently experiencing both menstruation and menopause). In addition, 69% of respondents reported experiencing a gynaecological health condition. Of those respondents, the most frequently reported health conditions were heavy menstrual bleeding (59%), menstrual irregularity (29%), endometriosis (15%), difficulties conceiving (12%), polycystic ovary syndrome (11%), miscarriage (11%), ovarian cysts (10%), fibroids (10%), incontinence (7%), and undiagnosed pelvic pain (6%). There was also a range of other less-frequently reported conditions cited, including interstitial cystitis, vulval pain, vaginitis, pelvic floor disorder, and prolapse. Many respondents reported comorbid gynaecological health conditions. 

The respondents’ experiences of menstrual health at work could be broken down into three more specific interrelated areas regarding periods, gynaecological health conditions, and menopause. The four main aspects of ‘blood work’ identified from the data were: 1. Managing the leaky, messy, painful body, 2. Managing access to facilities, 3. Managing stigma, and 4. Managing workload. In the following sections, we present the findings using data from respondents that are more widely emblematic of the data, and illustrative of the four aspects of bloodwork that we identify. 

### 4.1. Managing the Leaky, Messy, Painful Body

The data revealed that respondents reported considerable time and effort being spent in managing their menstrual symptoms at work, in particular, managing leaks, messiness, and pain. The respondents reported having strategies to manage pain including using prescription medication. Overall, respondents reported that they continued to work through the pain, even though they perceived it to have a debilitating effect on their ability to function. Several gynaecological health conditions and menopause are associated with unpredictable menstruation. For respondents, this unpredictability was difficult to manage alongside academic work across times and spaces. The participants also commented on how their menstrual symptoms could result in symptoms of fatigue:


*“Cramps can be very bad at night, leaving me really fatigued by day. Generally, by day I’m OK, though I get more tired quickly from teaching.”*
(Lecturer, Social Sciences, Northern Ireland)

Alongside managing their pain and fatigue, the data also showed how respondents had concerns regarding the appearance of menstrual blood and accidentally leaking onto chairs, with this fear becoming a reality for several respondents. These concerns were exacerbated for respondents when working away from familiar offices, where they were able to keep menstrual products to hand. The participants described having to resort to using toilet roll in lieu of reliable and comfortable products:


*“My period has started unexpectedly during conferences when I haven’t had any sanitary products with me... I’ve had to resort to stuffing toilet paper in my pants. This made me very anxious the whole time because I was worried about leaks.”*
(Assistant Professor, Full-Time Open-Ended contract, Social Sciences, Wales)

Other sensory ‘clues’ caused concerns for the respondents who interacted with their work environment differently to ensure that their menstruation was hidden from others:


*“I don’t walk between the students... I’m scared my menstruation blood and scent is too strong.”*
(Research Assistant, Part Time and Temporary Contract, Humanities, England)

The above research assistant, as well as expressing concerns that she could be identified and shamed for smelling different when on her period, also reflected on her changes in mood and behaviour as part of her menstrual cycle. These psychological symptoms were common across the dataset, where descriptions of managing pre-menstrual syndrome (PMS) in the workplace resembled wider narratives relating to the stigma of managing mental health conditions at work:


*“I have experienced severe PMS. This manifests in a variety of ways—depression, anxiety, social isolation, feelings of social rejection, irrational anger, feeling overwhelmed/unable to cope with small issues, insomnia, lack of motivation, self-doubt, negative thoughts and suicidal thoughts/thoughts of self-harm. Some days I could barely get out of bed or leave my house. I doubted everything I would do at work, things would take me longer, I wouldn’t want to be around people. meetings were extremely hard. My workload felt completely impossible... ”*
(Research Assistant, Full-Time Open-Ended Contract, Social Sciences, Scotland)

This participant outlined how the tasks and responsibilities of her role conflicted with her menstrual symptoms. The participants’ management of their physical and psychological symptoms highlighted how it was incumbent on them as individuals to manage their bodies at work. Managing the pain and risk of leaking for the above participants required access to specific products, including pain relief, menstrual products, and adequate toilet facilities, itself being identified as a public health concern [4]. The lack of access to menstrual products on university campuses or at conferences was reported as a significant cause of distress, discomfort, and disruption, and it is discussed in the following section.

### 4.2. Access to Facilities

Concerns over blood work were exacerbated by the apparent lack of necessary facilities at 7% of respondents’ employing universities (particularly sanitary product disposal bins). Less than 2% of respondents reported that their employers provided support for menstruation in the form of sanitary towels, tampons, or other products. Other recurring themes for this aspect of blood work included difficulty in accessing toilets to change menstrual products and to manage continence issues that are linked to gynaecological health conditions. Respondents reported a need to access toilet facilities very regularly—every 20 minutes, for example—due to ‘flooding’ from heavy uterine bleeding. This was seen to be very disruptive in meetings, but it was of particular concern within teaching, and respondents felt that students might react badly and perceive the lecturer to lack professionalism. Hence, the need to be able to access toilets could be very disruptive. The following quote is particularly illustrative of how respondents altered their working practices to accommodate their menstrual symptoms:


*“Heavy bleeding means that I need to leave meetings for breaks before they are over, that I plan my days around access to toilets, not having a private toilet for adequate washing... needing to leave teaching situations in order to change sanitary wear.”*
(Librarian, Part-Time, Open-Ended Contract, England)

The above respondent reorganised their time and movement around the university environment in comparison to when they are not menstruating. Reflecting wider research relating to managing various health conditions [58], access to toilets emerged as key for respondents, with many reporting that they had to plan their day around when they would be able to access toilets. The organisation of space at work, particularly in teaching rooms, was also seen by many respondents to be inadequate in not allowing privacy, particularly for those respondents managing heavy uterine bleeding. This could lead to considerable disruption, including the need to take sick leave or negotiate workplace adjustments which are usually reserved for disabled employees or those with long-term health conditions—although with notably limited application [76]. This is highlighted in the following quote, where a respondent describes the ways in which she had to draw on sick leave and negotiate an alteration in the teaching arrangements:


*“I had to call in sick because I was leaking through dressings at such a fast rate that I could not teach in the space we were using, with students surrounding the instructor on all sides, including behind. I insisted that I needed a different set-up for a number of reasons and was successful in obtaining a more appropriate and comfortable lecturing space.”*
(Associate Professor, Full-Time Open-ended Contract, Art and Design, England)

The data presented so far highlight how, for those managing their menstrual cycle at work, the issues they face replicate those of people with other health conditions. Although the menstrual cycle is not an illness and does not have a diagnosis or prognosis pathway, it can involve the individual managing both physical and psychological symptoms, the control and concealment of bodily fluids, and access to specific facilities to avoid the ‘exposure’ of their menstruating status and feeling shame and stigma. 

### 4.3. Managing Stigma

As has been touched upon already, a feature of the bloodwork of participants was attempting to avoid feelings of shame and being stigmatised. 63% of participants explained that they opted not to discuss menstruation or gynaecological health at work. Many participants reported feeling that they could not discuss their gynaecological health at work. The reasons for this varied. For some, menstruation or gynaecological health conditions were not something that ‘belongs in the workplace’ or were too personal to discuss at work. For others, a culture of precarity and research performance management was reported as making any form of ill health or menstrual difficulty impossible to discuss for fear of being rejected for employment. These concerns are expressed clearly in the following quote:


*“In an academic context, I worry that mentioning menstruation problems could be seen as a sign of weakness, and with everything being so precarious and competitive, talking about problems; whether that’s period pain, personal issues, or mental health problems, could put you in the “no” pile.”*
(PhD Student, Humanities, England)

Respondents’ fear of the stigma of menstruation was a strong theme and seen to be a by-product of the male-dominated nature of universities. The respondents had concerns about colleagues and students being aware of them menstruating, either through leaking (as discussed in an earlier section) or through the exhibiting of pain. The quote that is presented below reveals the internalised pressure to manage pain in a way that leads to presenteeism:


*“Due to the shame and stigma around menstruation, though, I often have to be physically present for work activities even if I am unable to carry them out.”*
(PhD Student, Part Time Temporary University Teacher, Humanities, England)

The stigma of menstruation and gynaecological health more broadly was associated with additional labour from respondents who reported that they changed their normal day-to-day activities at work in order to protect themselves and others from the sight or smell of menstrual blood. These changes could cause considerable disruption to their daily working lives. The perceived stigma that is associated with the (mis)management of menstrual blood highlights how blood work is not necessarily limited to the person who is menstruating. Embarrassment, shame, and guilt are social emotions, therefore blood work may extend to sympathetic colleagues, depending on the situation. 

In addition to the stigma of menstrual blood, psychological symptoms of premenstrual syndrome (PMS) were also perceived to be highly stigmatised in the academic workplace. As the respondent below reports, severe PMS can have a debilitating effect on the ability to work productively, although it is dismissed and laughed at when at work: 


*“Severe PMS is equally as challenging to manage as physical symptoms and yet can be dismissed as being ‘a bit moody’ or ‘we all get mood swings’. There is a lot of stigma around it and everything seems to be focused on supporting those with physical symptoms. PMS is seen as a joke. I am lucky to have a team that don’t think that way and understand that it is just as serious an issue”*
(Research Assistant, Full-Time Open-Ended Contract, Social Sciences, Scotland)

The same respondent continued, describing the importance of supportive colleagues in recognising the seriousness of the effects of PMS and the positive difference that is made by having a line manager who was well informed and prepared to put in place workplace adjustments. The extent to which respondents reported stigma was influenced by the gendered composition of the team that they worked for, with those working in female-dominated teams reporting greater freedom to discuss menstruation at work. As the respondent below revealed, there was a perception that both men and women can feel awkward about discussing menstruation: 


*“Depends on the colleague—I have definitely discussed having endometriosis with female colleagues. I have previously told female managers that I was going home due to period pain... Male colleagues, I’d probably feel less comfortable talking to—potential for awkwardness on both sides. On the whole, there is definitely still a lot of stigma around menstruation in the workplace, and I don’t think it’s widely discussed.”*
(Assistant Professor, Full-Time Open-Ended Contract, Social Sciences, Australia)

Some of the respondents reported that they did not understand their own internalised stigma in relation to menstruation, while others reported they purposefully discussed menstruation with their colleagues in order to challenge their own and others’ menstrual stigma. This stigma was particularly hard for trans and non-binary respondents, who indicated gender dysphoria (felt sex and gender identity discrepancy) was exacerbated during menstruation, making it harder to undertake activities such as lectures, and to discuss menstruation work.

Menstruation and living with gynaecological health conditions were often seen by respondents as incompatible with an academic career, with silence being maintained by respondents to preserve a sense of professionalism. Though a small number of respondents indicated that they tried to discuss menstruation and gynaecological health at work, as they were committed to overcoming stigma and making menstruation a topic of acceptable conversation within the workplace; in the context of precarity, which can dominate UK academics’ employment statuses, pain, bleeding, or any form of ill health was seen to be in direct conflict with employability.

### 4.4. Managing the Blood-Workload

Most of the respondents did not feel that menstruation had implications for their day-to-day working lives when asked in the broadest terms. Crucially, however, 36% did report daily difficulties that are associated with menstruation. Where these difficulties involved managing menstrual pain and heavy bleeding that interfered with work or created problematic additional demands, they reflected what this paper terms blood work. In many cases of general menstruation, the experiences typically lasted a few days each month. Here, a reasonable provision of home working from an employer could make a significant difference, and it was seen to set academia apart from other sectors, as is highlighted in the following quote:


*“Pre-PhD I worked in a non-academic environment where I was not allowed to work from home at all and found it even more difficult to manage. I value the ability to work from home during the first couple of days of at least some of my periods.”*
(Senior Lecturer/Associate Professor, Humanities, England)

For other participants, even with pain medication, it was not possible to work at all in the early days of their period, but they felt pressure to attend work, despite being unable to meaningfully engage. As the respondent below suggests, stigma that is associated with menstruation and pressure to attend work even when incapacitated by pain and heavy bleeding could result in presenteeism.

Despite evidence that academia provides flexibility to work from home, several respondents reflected on the perceived inflexibility of academic workloads to accommodate the variations in pain and levels of fatigue that are experienced by academics with gynaecological health conditions. One stark example below, of a woman who experienced a miscarriage and unsympathetic male colleagues, illustrates the inflexibility of the organisation of work and the perceived limits of organisational support:


*“I’ve witnessed and supported female colleagues go through bloody awful miscarriage experiences (one stuck on a research boat at sea with only men onboard when she had to persuade her project director to arrange a boat to get her to shore before the end of the voyage). None of these are taken into account in terms of workload, REF [Research Excellence Framework], CPD [Continuing Professional Development], let alone in terms of actual support. It’s just part of what women are expected to handle—as well as all the pressures of the profession. With babies, pregnancy, miscarriage and breast-feeding, one colleague estimates she was hormonally altered/pregnant etc for 6 years. I reckon it was five for me. I got a special chair when I was pregnant with twins—that’s the sum total of work-based support.”*
(Senior Research Fellow, Social Sciences, England)

This quote illustrates the additional labour that is required of those managing their gynaecological health conditions at work. This respondent illustrates a concern expressed by a number of respondents—that this additional labour is not incorporated into the targets within academic careers, and that a worker who experiences bodily phenomena such as miscarriage is required to individually manage this additional labour because workplaces and the organisation of work are implicitly designed for those who do not menstruate. In the following section, we discuss how responding to the biopsychosocial phenomena of menstruation and gynaecological health conditions in the workplace at a structural and organisational level can minimise the additional labour of women and trans/non-binary people at work and create more equitable, healthy workplaces.

## 5. Discussion 

This paper addresses an important gap in public and occupational health, as well as organisational studies: the experience of menstruation and gynaecological health conditions in the workplace [2]. Menstruation is uniquely situated as a biopsychosocial experience for women and trans/non-binary people, which is not an illness, but shares many facets of ill-health including physical and psychological symptoms, stigma and the need to access specific resources. We have termed the additional labour that is associated with managing menstruation and associated gynaecological health conditions in the workplace as blood work. This theorisation draws on both notions of body work [42] and body work with vaginas and vulvas [43]. It builds on and extends research that has focused on work done by workers on the bodies of others (e.g., clients or patients). 

The presented data reveal the considerable challenges that are faced by those managing gynaecological health conditions and menstruation while navigating employment. The respondents reported a range of difficulties that are associated with combining their gynaecological health with idealised notions of academia in managerialist universities preoccupied with performance targets that are already known to negative health outcomes, disproportionately borne by women [71]. More specifically, the organisation of work and the physical environment, along with internalised stigma and the material reality of a painful and leaking body, combined to create additional labour for participants. We call this labour ‘blood work’ and have analysed it in terms of four main inter-related facets (see Table 1).

This delineation of blood work resonates with Warhust and Nickson’s [40] concept of aesthetic labour, which explores the work that was done by workers to change their bodies for the benefit of their employers. The current research differs, in that it has identified the work done by workers on their own bodies, to shield others from their menstruation and gynaecological health conditions and maintain gendered norms of professionalism. 

It is clear from the data that combining work as an academic with experiencing menstruation requires considerable additional labour. Those academics who experienced menstruation and associated gynaecological health conditions reported additional labour in relation to academic workloads and managerialist/neoliberal performance management systems in UK universities, managing paid work with leaking menstrual blood, stigma, and navigations of the physical working environment. Studies of maternal body work, such as breastfeeding, come closer to understanding how women and non-binary people manage the pregnant body in the workplace to adhere to the norms of disembodied workers [32]. However, this work does not fully capture the management of specific stigma relating to blood, menstruation, and gynaecological health. Equally, while aesthetic labour does allow for an understanding of the body and the work an employee may do on their own body for an employer’s gain [40], it does not adequately capture experiences of managing pain, blood, the physical working environment, or norms of the ideal (academic) worker. 

Previous research has pointed to menstrual etiquette, whereby all aspects of menstruation must be concealed [33], which does capture some of the concealment aspects of how menstruation is managed at work, but does not extend to the more physical symptoms that were described by respondents. Analysis of the data showed particular forms of work that are not fully captured by these extant frameworks and, therefore, we propose a new term, which builds on ideas of body work, dirty work, and aesthetic labour to theorise four interconnected components of blood work—managing workloads, the leaky/messy/painful body, stigma, and (lack) of access to facilities (Table 1).

The difficulties that are faced by those who are not easily able to meet the expectations of universities, particularly the uninterrupted productivity to meet managerialist targets, are well established [47]. Regarding gynaecological health, difficulties may be particularly acute in the first days of menstruation, but can persist especially for those managing chronic gynaecological conditions. While previous research has captured many of the gendered experiences of women at work, including in academia, the menstruating body has, until now, been absent from such debates. The current study adds to this debate, illustrating the workplace effects on those managing menstruation in environments that do not accommodate women and trans/non-binary people’s bodies, and giving a voice to individual’s embodied workplace experiences. 

The study and data also point to the specific interplay between material realities of menstruation, such as blood and odour, alongside social, interrelational experiences, such as managing, or anticipating stigma. The ideal body at work is one that does not smell, as articulated by Riach and Warren [35]. This lack of discernible odour extends to menstruation, whereby the concealment of menstrual odour and dried blood was a prime concern. The participants were keen to conceal these odours from colleagues and from students, extending to the physical organisation of the teaching space and toilet facilities, representing an additional layer of blood work labour. This additional labour was more difficult in contexts where there was little access to suitable facilities and destigmatising colleague interactions, showing the interplay of the four components of blood work. 

While the study includes a considerable sample size covering an array of gynaecological health conditions, future research should consider the effects of particular conditions (e.g., endometriosis) on employment specifically, and the complex interactions between accessing healthcare, diagnosis, and workplace adjustments. Such work may wish to overcome the limitations that are inherent in cross-sectional research by adopting longitudinal designs that can follow experiences over time. Doing so would allow for these complexities to be mapped. We would also suggest that future work should move beyond the comparatively privileged workplace of academia, where employees are sometimes able to exploit the inherent flexibility in order to manage symptoms. Workplaces without such flexibility, where the body has been demonstrated to be of key importance to organisational goals and profitability, e.g., service work [40] would offer fruitful avenues for further workplace menstruation research. Additionally, given the important role of trade unions in facilitating the employment of disabled people, and those with chronic health conditions [77], future research may wish to explore the extent to which trade union representatives and other stakeholders are able to support employees with complex and stigmatised medical conditions.

The sample did include some trans and non-binary participants, and their distinctive experiences pointed toward considerable further challenges in the workplace, such as the exacerbation of gender dysphoria. Future research should consider prioritizing the voices of trans and non-binary people in order to understand how gender identity, workplace norms, and gynaecological health interact to inform experiences and employment outcomes. We recommend that further research explore, in detail, the potential intersecting work-related disadvantages that impact on the experiences of bloodwork. We would suggest this include research with those that are involved with physical work, including shift work, those on low incomes, and those that are required to wear and/or use specialist equipment. 

A key recommendation from this paper is for employers and policy makers. We suggest responding to workplace menstruation and gynaecological health conditions as a public health concern. We suggest “upstream” [78] policy and environmental alterations to workplaces that can minimise the individual’s need to complete bloodwork and, instead, refocuses the responsibility for this management to employers. Changes to ensure that adequate facilities are available and environments that recognise a substantial proportion of the workforce will menstruate will mitigate some of the gendered disadvantages women and trans/non-binary people experience in the workplace. 

## 6. Conclusions

In conclusion, menstruation and gynaecological health conditions represent a notable and unfortunate absence in general discussions of work, employment and public health. In this paper, data from a qualitative survey of university workers (predominantly academics) revealed the considerable additional labour that is required of those managing menstruation and gynaecological health conditions in the workplace. Existing theories of the body at work cannot fully explain these experiences. The additional labour experiences of menstruating employees involved managing high workloads, pain and leakage of menstrual blood, stigma, and access to necessary facilities. We have named this additional labour as blood work, in the hope that this naming can be used to reflect the specific work that is required in managing menstruation, menstrual blood, pain, and associated conditions. 

## Figures and Tables

**Table 1 ijerph-18-01951-t001:** Four facets of blood work undertaken by menstruating employees.

Facets of Blood Work	Examples
Managing workload	Difficulty with pain and managing teaching during first few days of menstruation, inflexible attendance expectations and presenteeism, managing performance targets
Managing the leaky, messy, painful body	Fatigue, leakage of menstrual blood on clothes and chairs, working through pain
Managing stigma	Shame, concealment of menstruation/purposeful discussion of menstruation to challenge stigma, fear of detection e.g., smell, fear of being laughed at
Managing (lack of) access to facilities	Insufficient toilet facilities (especially accessible from teaching spaces), lack of private access to toilets/disposal bins, no universal provision of menstrual products, arrangement of teaching spaces e.g., where staff member is surrounded by students who may detect menstruation (leakage/odour)

## Data Availability

The data presented in this study are available on request from the corresponding author. The data are not publicly available due to their sensitive content.

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
