# Peer review of "Blood Work: Managing Menstruation, Menopause and Gynaecological Health Conditions in the Workplace"

_ijerph, 2021, doi:10.3390/ijerph18041951_

Round 1

Reviewer 1 Report

Thank you for revising the paper according to the comment.

Author Response

Thank you for your positive feedback - we have proof read the paper 

Reviewer 2 Report

Looks good! Needs a line edit -- several typos in the new sections. Numbering of references in one out from 20 (see Jack 2019).

Author Response

Thank you for your kind comments, we have proof read the paper and resolved the referencing issue

This manuscript is a resubmission of an earlier submission. The following is a list of the peer review reports and author responses from that submission.

Round 1

Reviewer 1 Report

Thank you for an interesting and important paper on a significant and neglected area of study. I have three points to make that in my view would make the paper stronger. 

  1. Methodology justification: as you say, when dealing with a public health topic, survey data can be more readily collected, and you include this and other advantages. The paper would be enhanced by a brief reference also regarding your understanding of the limitations of survey data. Such as, interviews have the potential to elicit more nuanced and detailed data through conversational Q and A, and in some cases allow greater integration of contextual information into the analysis. See Jack et al 2018 on menopause in academic workers, and Owen 2018 on menstruation at work, both of which used interviews. Perhaps you could put a reference to your choice of method and what might be discovered if interviews were used in the future research section. 
  2. In the last line of the first paragraph of the introduction, "....organisation studies on menstruation as one of the three Ms of gendered working lives (the other two being maternity and menopause [2]" suggest a change to "....organisation studies on menstruation, conceptualised as one of the three Ms of gendered working lives (the other two being maternity and menopause [2]"
  3. I was surprised you did not include a reference to a research on menstruation in the workplace and specifically on the Coexist menstrual workplace policy mentioned in the first sentence of their article. The main author attended a paper on this in early 2018, and that article on the topic has been available since late 2018. In the context of research on menstruation, given the paucity of extant literature on the topic, other authors have included conference papers in their references but I will leave that up to you. The published paper should be referenced though.
  4. Quotes on page 10 need to be italicised 

Owen, L. (2018). Menstruation and humanistic management at work: The development and implementation of a menstrual workplace policy. Journal of the Association for Management Education & Development, 25(4), 23-31.

Owen, L. (2018). Paper: Menstrual materiality: Changing the permission field in the workplace. Menstruation Symposium, 17/1/18, University of St. Andrews. 

Author Response

Thank you for an interesting and important paper on a significant and neglected area of study. I have three points to make that in my view would make the paper stronger.  

  1. Methodology justification: as you say, when dealing with a public health topic, survey data can be more readily collected, and you include this and other advantages. The paper would be enhanced by a brief reference also regarding your understanding of the limitations of survey data. Such as, interviews have the potential to elicit more nuanced and detailed data through conversational Q and A, and in some cases allow greater integration of contextual information into the analysis. See Jack et al 2018 on menopause in academic workers, and Owen 2018 on menstruation at work, both of which used interviews. Perhaps you could put a reference to your choice of method and what might be discovered if interviews were used in the future research section.  

We thank the reviewer for this suggestion and have added a further sentence on page 5 to expand on the limitations of survey data and existence of interview-based papers on the topic: ‘Online surveys are established tools for collecting data from participants living with chronic health conditions [72], despite limitations relating to access and potential brevity in responses [73]. Research exploring workplace experiences of menstruation and menopause have largely been interview-based to access more detailed data from fewer participants [74][1].’   

  1. In the last line of the first paragraph of the introduction, "....organisation studies on menstruation as one of the three Ms of gendered working lives (the other two being maternity and menopause [2]" suggest a change to "....organisation studies on menstruation, conceptualised as one of the three Ms of gendered working lives (the other two being maternity and menopause [2]" 

We have incorporated the suggested wording with thanks. 

  1. I was surprised you did not include a reference to a research on menstruation in the workplace and specifically on the Coexist menstrual workplace policy mentioned in the first sentence of their article. The main author attended a paper on this in early 2018, and that article on the topic has been available since late 2018. In the context of research on menstruation, given the paucity of extant literature on the topic, other authors have included conference papers in their references, but I will leave that up to you. The published paper should be referenced though. 

We thank you for noting this omission and apologise. We have revisited the background section and methods and referenced ‘Owen, L. (2018). Menstruation and humanistic management at work: The development and implementation of a menstrual workplace policy. Journal of the Association for Management Education & Development, 25(4), 23-31’ - this reference is [1] in text. 

  1. Quotes on page 10 need to be italicised  

We have corrected this error, with thanks.  

Reviewer 2 Report

Despite the importance of reproductive health for the menstrual problem, research on this subject is lacking. Therefore, I think this is an important study that can contribute to women's reproductive health. Please consider the following points to modify or add content.

The introduction is too long because it contains the background. Divide the background into separate paragraphs. Then describe previous studies on mesnstrual health and work, and what limitations were in those studies. For example, a cohort study of American nurses is the best predecessor to reproductive health research. I attach a link.

And, please describe in the introductory part why the subjects were targeted for highly educated people.

Please describe the method part in the order of study design, participant, data collection, and data analysis. And in the method section, please describe why you asked the subjects what questions about menstrual health. In this study, the subject, sample size, setting, etc. are well described, but the content of the research question is insufficient.

Author Response

Reviewer 2 

  1. Despite the importance of reproductive health for the menstrual problem, research on this subject is lacking. Therefore, I think this is an important study that can contribute to women's reproductive health. Please consider the following points to modify or add content. 

Thank you for this comment. 

  1. The introduction is too long because it contains the background. Divide the background into separate paragraphs. Then describe previous studies on menstrual health and work, and what limitations were in those studies. For example, a cohort study of American nurses is the best predecessor to reproductive health research. I attach a link. 

We have provided a further subtitle ‘Background’ to clarify the conclusion of a short introductory section. We have also shortened the background sections (pages 2-5). 

  1. And, please describe in the introductory part why the subjects were targeted for highly educated people. 

On page 5, we have added two sentences clarifying our reasons for recruiting white collar workers: ‘Despite the gendered forms of management in university settings being well understood (21), little is known about employees’ embodied experiences of working in Higher Education. Universities offer excellent insight into the management of menstruation in white collar jobs where there is a perception of work flexibility’. We have also added a sentence to the discussion section (page 13) recommending further research be undertaken in this area to better understand intersecting workplace disadvantages and work types and their impact on Bloodwork: ‘We recommend that further research explore in detail the potential intersecting work-related disadvantages that impact on experiences of bloodwork. We would suggest this include research with those involved with physical work, including shift work, those on low incomes and those required to wear and/or use specialist equipment.’ 

  1. Please describe the method part in the order of study design, participant, data collection, and data analysis. And in the method section, please describe why you asked the subjects what questions about menstrual health. In this study, the subject, sample size, setting, etc. are well described, but the content of the research question is insufficient. 

We have reordered the methods section as per your suggestions (pages 5-6) and added further detail regarding the research questions in the survey [pg. 5-6]:  

‘The online survey asked participants to describe their experiences of menstruation, menopause and gynaecological health conditions at work. Open-ended questions were asked in relation to: managing menstruation, menopause and gynaecological health conditions at work, including the accessibility of facilities, the impact of managing menstruation in relation to teaching, research, administration, and conferences, taking leave, access to sanitary products and ability to discuss menstruation and gynaecological health at work. These questions gave respondents the opportunity to describe the work they engaged in to manage their menstrual and gynaecological health at work. Finally, participants were also asked to describe their gender, job title, discipline, location of work, contract type (e.g. full-time; open-ended). 

Reviewer 3 Report

Blood work: Managing menstruation, menopause and gynaecological health conditions in the workplace

This paper presents an overarching principle of blood work of menstruation as a factor affecting workers lives. The study follows the organization of a social sciences publication rather than a public health paper and could benefit from reorganization and greater focus.

Major comments

  1. The authors have been somewhat inclusive in identifying women and non-binary people who menstruate but in other places in the paper refer to women. It would be helpful to use a more inclusive term to identify people who menstruate as transmen who menstruate face even greater stigma and should be included in this nascent framework.

  1. The introduction is extremely long for papers in this field and could be considerably edited.

  1. Attention to low-wage settings and how affordability plays a role in blood work is absent.

  1. Results: The results are reported inconsistently and it is challenging to understand why certain quotes were selected? Were these emblematic of the type of responses that were seen or did these confirm what the researchers expected to find? How were the quotes selected?

Minor comments

  1. Line 32-33: What gap in the literature? This has not yet been established in the introduction.

  1. Line 95: This paragraph starts by mentioning economic hardships but does not provide this evidence in the rest of the paragraph.

  1. Lines 177-200: This section could be more focused on how menstruation relates to academia more specifically. It broadens discussion to general sexism in the workplace.

  1. Methods: When was the survey distributed? Was IRB approval sought?

  1. Methods, Sample: This section is reporting results and should be moved to the results section.

  1. Formatting inconsistencies in results under the “Managing the blood-workload” section.

Author Response

Reviewer 3 

This paper presents an overarching principle of blood work of menstruation as a factor affecting workers lives. The study follows the organization of a social sciences publication rather than a public health paper and could benefit from reorganization and greater focus. 

 Thank you for this comment. The authors are all exclusively social scientists, so your support in shaping the paper for a public health audience is very welcome.  

Major comments 

  1. The authors have been somewhat inclusive in identifying women and non-binary people who menstruate but in other places in the paper refer to women. It would be helpful to use a more inclusive term to identify people who menstruate as transmen who menstruate face even greater stigma and should be included in this nascent framework. 

We have ensured that language relating to the current study is inclusive. When referring to the work of other academics and researchers, we use language that is reflective of theirs. We have made this clear on page 2, in the introduction: ‘Though the authors of this article discuss people of any gender identity who experience menstruation, in discussing existing research we default to the language used in those studies.’ 

The use of more inclusive language has been incorporated throughout to provide greater consistency. However, as is now stated at the bottom of page 2, though our research acknowledges and recognises the experiences of people of any gender identity experiencing menstruation, when discussing previous work, we use the language of the authors whose work we are citing. In our discussion we also suggest further research be conducted to explore the experiences of trans and non-binary people [pg.13]: The sample did include some trans and non-binary participants, and their distinctive experiences pointed toward considerable further challenges in the workplace, such as the exacerbation of gender dysphoria. Future research should consider prioritizing the voices of trans and non-binary people in order to understand how gender identity, workplace norms and gynaecological health interact to inform experiences and employment outcomes.’ 

  1. The introduction is extremely long for papers in this field and could be considerably edited. 

 To provide clarity regarding the conclusion of the introduction we have added the additional subtitle of ‘background’, which is the most substantive section of the paper other than the findings section. We have shortened the background sections removing paragraphs from each section. 

  1. Attention to low-wage settings and how affordability plays a role in blood work is absent. 

We did not ask our participants about their income, and as such are unable to make comment about whether affordability affects bloodwork. We have added a sentence to the discussion section (page 13) recommending further research be undertaken in this area to better understand intersecting workplace disadvantages and work types and their impact on Bloodwork: ‘We recommend that further research explore in detail the potential intersecting work-related disadvantages that impact on experiences of bloodwork. We would suggest this include research with those involved with physical work, including shift work, those on low incomes and those required to wear and/or use specialist equipment.’ 

  1. Results: The results are reported inconsistently, and it is challenging to understand why certain quotes were selected? Were these emblematic of the type of responses that were seen or did these confirm what the researchers expected to find? How were the quotes selected? 

 To address this suggestion, we have supplied a more specific introduction to the findings section [pg. 6-7]: The respondents’ experiences of menstrual health at work could be broken down into three more specific interrelated areas regarding periods, gynaecological health conditions and menopause. The four main aspects of ‘blood work’ identified from the data were - 1. Managing the leaky, messy, painful body, 2. Managing access to facilities, 3. Managing stigma, and 4. Managing workload. In the following sections we present the findings using data from the interviews that are emblematic of the data more widely, and illustrative of the 4 aspects of bloodwork we identify. 

Throughout the findings, we provide more clarity regarding the data [pg 7-11], have removed superfluous data and reordered some content. . 

Minor comments 

  1. Line 32-33: What gap in the literature? This has not yet been established in the introduction. 

We have further clarified in the introduction: ‘Such examples of changes in policy point to what is otherwise a strikingly absent topic in studies of occupational health and organisation studies on menstruation, conceptualised as one of the three Ms of gendered working lives (the other two being maternity and menopause [2]) ...The current paper addresses this gap in understandings of how people manage menstruation and associated gynaecological health conditions in the workplace 

  1. Line 95: This paragraph starts by mentioning economic hardships but does not provide this evidence in the rest of the paragraph. 

 Our evidence related to economic hardship is provided in preceding paragraph and wording has been altered to clarify. 

  1. Lines 177-200: This section could be more focused on how menstruation relates to academia more specifically. It broadens discussion to general sexism in the workplace. 

 We have added further detail in the background section to maintain focus on university work: ‘This important research explores cultural associations between menstruating and impurity and the resulting limited access to public spaces for women and trans/non-binary people and girls [18]. In low to middle income countries, it has become apparent that data on how women manage menstruation and work are very sparse, complicated by women’s predominant employment in informal sectors where there are fewer legal protections [19]. University based work offers excellent insight into the management of menstruation in white collar jobs where there is a perception of work flexibility, an area under explored within wider literature on menstruation and employment.’ [pg. 2] 

  1. Methods: When was the survey distributed? Was IRB approval sought? 

We have added the distribution year of the survey, 2017. On page 6 we state: ‘The study received full ethical approval from the lead author’s institution. Survey respondents were informed of their right to withdraw, and that their responses would be anonymised and not shared with a third party. As such, the participants were fully informed of the purpose of the study.’ 

  1. Methods, Sample: This section is reporting results and should be moved to the results section. 

We have reordered our Methods sample as per Reviewer 2’s suggestion and moved details of respondents relevant characteristics to the findings section as per reviewer 3’s suggestion.  

  1. Formatting inconsistencies in results under the “Managing the blood-workload” section. 

We have corrected formatting inconsistences and typos in this section.  

We would like to thank the reviewers again for their comments. As a result of this thoughtful and considered engagement we have made substantial revisions to the attached paper. We feel that the paper is clearer. We appreciate the opportunity to revise and resubmit and would welcome any further guidance on the paper